

# Incidence and risk factors of new-onset sacroiliac joint pain after spinal surgery: a systematic review and meta-analysis

ChengHan Xu[1,2], Xuxin Lin[1,2], Yingjie Zhou[1], Hanjie Zhuo[1], Lei Yang[1], Xubin Chai[1] and Yong Huang[1]

[1] Luoyang Orthopedic Traumatological Hospital of Henan Province (Henan Provincial Orthopedic Hospital), Luoyang, Henan Province, China
[2] Hunan University of Chinese Medicine, Changsha, Hunan Province, China

## ABSTRACT

**Purpose:** A systematic review and meta-analysis for incidence and risk factors of new-onset sacroiliac joint pain (SIJP) after spinal surgery aimed to provide evidence-based medical references for its early prevention, timely intervention, and appropriate treatment.

**Methodology:** The protocol of the systematic review and meta-analysis was registered in the International Prospective Register of Systematic Review (PROSPERO) with the PROSPERO ID (CRD42023463177). Relevant studies were searched to January 2024 from the databases of PubMed, Embase, Cochrane Library, and Web of Science, and the types of studies were cohort studies, case-control studies, and cross-sectional studies. Study quality was assessed using the Newcastle-Ottawa Scale (NOS) and the Cross-Sectional/Prevalence Study Quality recommended by the Agency for Healthcare Research and Quality (AHRQ). Two authors conducted studies search, data extraction, and quality assessment independently. Meta-analyses were done using Stata 14.0 software.

**Results:** Twelve observational studies with 3,570 spinal surgery patients were included. Ten were case-control studies, one was a cross-sectional study, and another was a cohort study, all of which were of moderate quality and above. The results of the meta-analysis showed that the incidence of new-onset SIJP after spinal surgery was 9.40%; females, no. of surgical segments, fusion to the sacrum, and postoperative pelvic tilt (PT) were significantly associated with the new-onset SIJP after spinal surgery. Meta-analyses for preoperative and postoperative controls of spondylopelvic parameters showed that postoperative lumbar lordosis (LL) in the SIJP group and postoperative LL and sacral slope (SS) of patients in the NoSIJP group had significant differences from preoperative.

**Conclusion:** Available evidence suggests that an increased risk of new-onset SIJP after spinal surgery is associated with sex, multi-segmental surgery, fusion to the sacrum, and greater postoperative PT.

Corresponding author
Yingjie Zhou, 1099168230@qq.com

## INTRODUCTION

Chronic lower back pain that persists or newly emerges after spinal surgery is a formidable challenge for spinal surgeons, significantly affecting the postoperative quality of life for patients as well as posing a socioeconomic burden (*Elsamadicy et al., 2017*; *Inoue et al., 2017*). Sacroiliac joint pain (SIJP) is one of the significant contributors to lower back pain after spinal surgery (*Maigne & Planchon, 2005*; *DePalma, Ketchum & Saullo, 2011*; *Liliang et al., 2011*; *Yoshihara, 2012*). However, due to the similarity in pain patterns with chronic pain pathological conditions such as lumbar spine degeneration or adjacent segment disease after spinal surgery (*Deer et al., 2021*), the diagnosis and treatment of SIJP are particularly challenging (*Chuang et al., 2019*).

The SIJ is typically a diarthrodial synovial joint that is mechanically most stable over the spinal-pelvic region, only with minimal rotational and translational motion (*Goode et al., 2008*). The SIJ is also the largest axial joint in the human body and acts as a crucial shock absorber between the spine and the lower limbs; it not only transmits and dissipates axial compressive and rotational stresses but also withstands medially directed forces better than the lumbar spine, performing an essential biomechanical function in physiological activities (*Vleeming et al., 2012*; *Cohen, Chen & Neufeld, 2013*). Despite the relative stability of the SIJ, the causes of new-onset SIJP may be various due to the complex anatomy of the SIJ (*Kiapour et al., 2020*) and the rich innervation of its surrounding ligaments (*Cohen, Chen & Neufeld, 2013*), such as a history of acute and overuse injuries to the pelvic girdle and lower limbs (*Abdollahi et al., 2023*), pregnancy (*Fiani et al., 2021*), athletes involved in partially unilaterally loaded sports (*Kato et al., 2022*), low body mass index (*DePalma, Ketchum & Saullo, 2012*).

Spinal surgery, especially lumbar interbody fusion (LIF), is also a significant cause of SIJP (*Yoshihara, 2012*). However, identifying the risk factors of new-onset SIJP after spinal surgery needs more evidence in light of the limited available studies and conflicting findings from existing studies on identical factors. This study conducted both qualitatively and quantitatively a comprehensive systematic review and meta-analysis of previous relevant studies to explore risk factors of new-onset SIJP after spinal surgery while providing evidence-based medical references for its early prevention, timely intervention, and appropriate treatment.

## METHODOLOGY

The design and implementation of this systematic review and meta-analysis followed the Preferred Reporting Items for Systematic Reviews and Meta-Analyses (PRISMA) guidelines 2020 (*Page et al., 2021*) with the PROSPERO ID of this study protocol (CRD42023463177).

### Data sources and search strategy

The data sources of this study mainly were from studies on new-onset SIJP after spinal surgery searched up to January 2024 in the databases PubMed, Embase, Cochrane Library, and Web of Science. The search strategy consists of a combination of Medical Subject Headings (MeSH) terms such as Sacroiliac Joint, Pain, Spinal fusion, and relevant

Textwords terms. In addition, we manually screened references from relevant literature and previous systematic reviews to minimize the risk of omission.

## Inclusion and exclusion criteria

### Inclusion criteria

(1) Observational study: Cohort studies, case-control studies, or cross-sectional studies.
(2) The study population consisted of patients who had no preoperative symptomatic SIJP and underwent spinal surgery. The surgical site was predominantly lumbar, with or without thoracic involvement. The main types of surgery were decompression, LIF, or multi-segment corrective fusion, either open or mini-invasive surgery.
(3) The outcome indicator was whether or not patients had a new-onset of SIJP after spinal surgery. The diagnosis of SIJP required a combination of symptoms, physical examination, provocation tests, a diagnostic scoring system for SIJP (*Kurosawa et al., 2017*), and most critical diagnostic SIJ injection/block/infiltration with a positive response.
(4) Studies reported at least one or more of the relevant factors such as sex, age, preoperative diagnosis, No. of surgical segment, fusion to sacrum, and spondylopelvic parameters: lumbar lordosis (LL), pelvic incidence (PI), pelvic tilt (PT), and sacral slope (SS).

### Exclusion criteria

(1) Reviews, letters, comments, case reports, non-English studies, and non-human studies.
(2) The study populations included patients with preoperative SIJP or excluded patients without low back pain after spinal surgery.
(3) The same authors or institutions published different studies containing duplicated subjects.
(4) The data in studies was ambiguous or unextractable.

## Study selection and data extraction

Two authors (CH Xu and XX Lin) independently screened the retrieved articles according to the inclusion and exclusion criteria, with initial exclusion based on titles and abstracts followed by a full-text examination of potentially eligible articles. The final included studies reached a consensus by cross-verifying the articles screened by the two authors and resolving disagreements of inclusion through discussion with a third author (L Yang).

Data extraction from included studies encompassed basic information about the included studies, such as the name of the first author, year of publication, country, type of study, sample size, and diagnostic criteria for SIJP; baseline characteristics of the study subjects such as age, gender, et al.; and incidence for new-onset SIJP after spinal surgery and relevant factors. Two authors (CH Xu and XX Lin) independently extract.

## Study quality assessment

We assessed the quality of cohort and case-control studies using the Newcastle-Ottawa Quality Assessment Scale (NOS) out of a possible nine, with a score of seven and higher indicating high quality. The quality assessment of cross-sectional studies employed an 11-item checklist recommended by the Agency for Healthcare Research and Quality (AHRQ).

Each item scored according to the answer of "yes," "no," or "unclear," with a score of 1 if it was "yes" and 0 otherwise, and a total score of eight and higher indicated high quality. Two authors (CH Xu and XX Lin) scored independently, with disagreement resolved through discussion with a third author (L Yang).

## Statistical analysis

Meta-analyses were conducted with Stata 14.0 statistical software. The effect sizes for dichotomous variables were odds ratio (OR) and 95% confidence intervals (CIs), and for continuous variables, they were standardized mean difference (SMD) and 95% CI. SMD values between 0 and 0.2 indicated a low-risk factor, 0.2–0.5 as a moderate risk factor, 0.5–0.8 as a high-risk factor, and >0.8 as a very high-risk factor. Cochran Q chi-square test and $I^2$ statistic were used to assess inter-study heterogeneity. We performed meta-analyses using a fixed effect model for low inter-study heterogeneity ($I^2 < 50\%$ and Q test $P > 0.1$). Provided that high inter-study heterogeneity ($I^2 > 50\%$ or Q test $P < 0.1$), we would find the source of heterogeneity through subgroup or sensitivity analysis. If the source of heterogeneity failed to be identified, a random effects model was employed for meta-analyses. By sensitivity analysis to determine whether the results were stable, and if not, we abandoned the quantitative analysis using a random effects model in favor of a qualitative systematic review. Funnel plots with Begg's and Egger's tests were employed to assess publication bias among the studies only if there were more than or equal to ten studies. $P$-value <0.05 indicated statistically significant.

## RESULTS

### Literature search

We identified a total of 1,975 articles by a comprehensive search across various databases. After removing 720 duplicates and 397 publications such as reviews, meta-analyses, case reports, letters, or animal experimentation, 828 irrelevant studies were excluded based on titles and abstracts. Then, we screened the remaining 29 articles by carefully reading the complete text and finally included 12 in the systematic review. Some articles that did not meet the criteria, such as studies published by the same authors or the same institution in which there was the reuse of samples (*Shin et al., 2013*; *Unoki et al., 2017*), failure to reported at least one of the relevant factors (*Noureldine et al., 2021, 2023*), or ambiguous or unextractable data (*Abouzeid, 2016*; *Nessim et al., 2021*), were excluded (Fig. 1).

### Characteristics of included studies and quality assessment

There were 3,570 patients after spinal surgery among the 12 studies finally included (*Cho et al., 2013*; *Schomacher et al., 2015*; *Unoki et al., 2016*; *Guan et al., 2018*; *Lee, Lee & Harman, 2019*; *Tonosu et al., 2019*; *Unoki et al., 2019*; *Murata et al., 2022*; *Yan et al., 2022*; *Kalidindi et al., 2023*; *Yang et al., 2023*; *Xu et al., 2024*), of which 325 were SIJP patients. Table 1 shows the main characteristics and quality assessment of the studies.

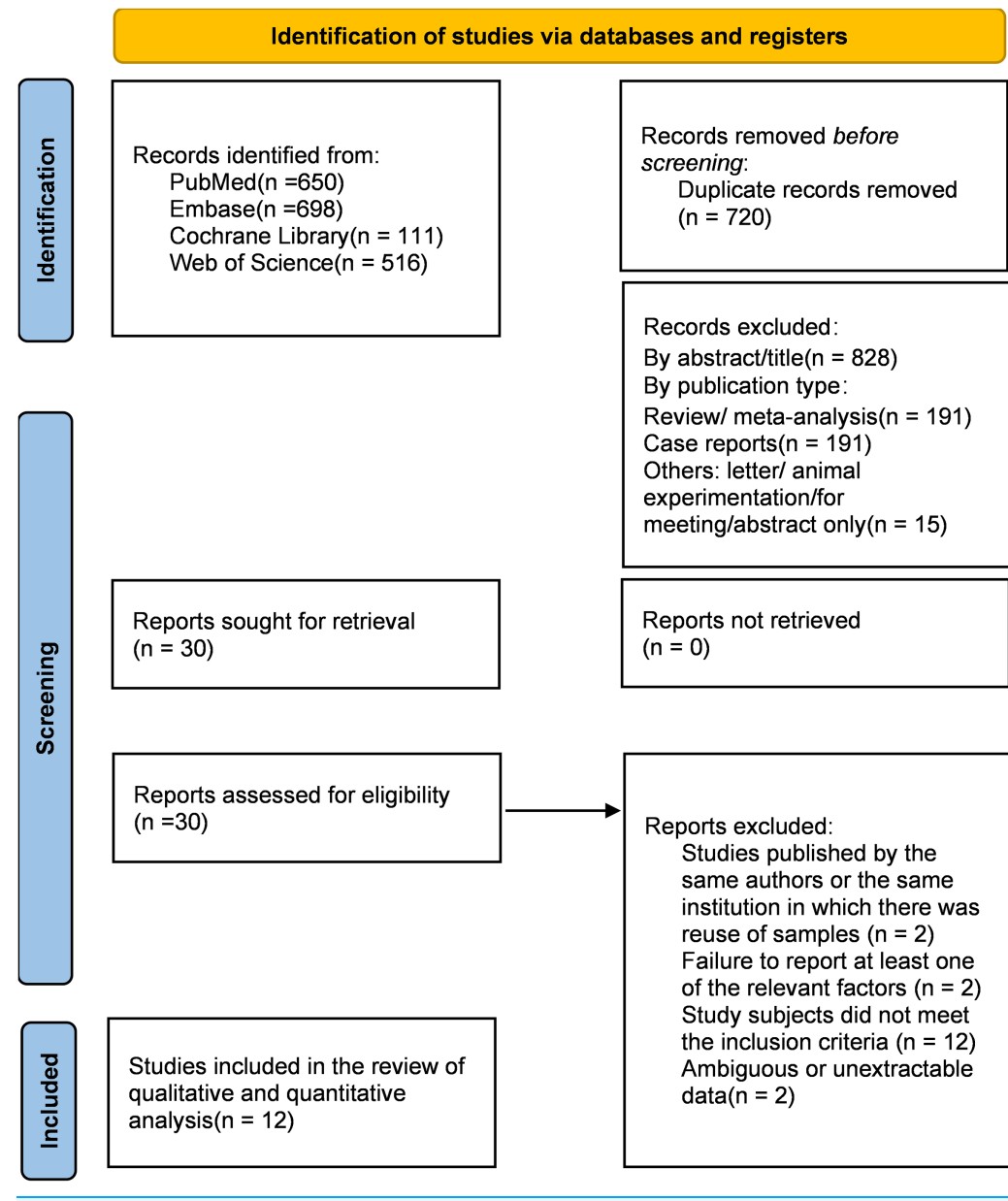

**Figure 1** **PRISMA 2020 flow diagram (*Page et al., 2021*) for screening the articles included in the meta-analysis.**

## Incidence of new-onset SIJP (Figs. 2 and 3)

A meta-analysis of 12 studies that reported the incidence of new-onset SIJP after spinal surgery had high inter-study heterogeneity ($I^2 = 84.3\%$, $P < 0.000$) (*Cho et al., 2013*; *Schomacher et al., 2015*; *Unoki et al., 2016*; *Guan et al., 2018*; *Lee, Lee & Harman, 2019*; *Tonosu et al., 2019*; *Unoki et al., 2019*; *Murata et al., 2022*; *Yan et al., 2022*; *Kalidindi et al., 2023*; *Yang et al., 2023*; *Xu et al., 2024*). We performed subgroup analyses according to the type of surgery and studies' publication time, ethnicity, continent, type, and sample size, respectively, but failed to find the source of heterogeneity. The pooled results of a random

**Table 1 Characteristics of included studies and quality assessment.**

| First author | Year | Nation | Research type | Type of surgery | Definition of SIJP | Sample Size | Patients with SIJP | NOS |
|---|---|---|---|---|---|---|---|---|
| *Cho et al. (2013)* | 2013 | Korea | Retrospective | Posterior LIF | ①④ | 452 | 28 | 8 |
| *Schomacher et al. (2015)* | 2015 | Germany | Retrospective | Decompression | ②④ | 100 | 22 | 7 |
| *Unoki et al. (2016)* | 2016 | Japan | Retrospective | Lumbar fusion surgery | ①②④ | 262 | 28 | 9 |
| *Guan et al. (2018)* | 2017 | China | Retrospective | Lumbar open diskectomy (40.9%)/posterior LIF (59.1%) | ①②④ | 472 | 65 | 7 |
| *Unoki et al. (2019)* | 2019 | Japan | Retrospective | Multi-segment corrective fusion ≥3 | ③④ | 77 | 12 | 7 |
| *Tonosu et al. (2019)* | 2019 | Japan | Prospective | Anterior and posterior lumbar spine surgeries (fixation surgery 34.0%) | ①③④ | 265 | 8 | 8 |
| *Lee, Lee & Harman (2019)* | 2019 | UK | Retrospective | Lumbar fusion surgery | ①②④ | 317 | 38 | 6 |
| *Murata et al. (2022)* | 2022 | Japan | Prospective | Long corrective fusion with lumbosacral posterior LIF | ③④ | 94 | 11 | 8 |
| *Yan et al. (2022)* | 2022 | China | Retrospective | Posterior thoracolumbar fusion | ①②③④ | 409 | 23 | 6* |
| *Kalidindi et al. (2023)* | 2023 | India | Retrospective | Transforaminal LIF involving L4-L5/L5-S1 | ①②④ | 354 | 34 | 8 |
| *Yang et al. (2023)* | 2023 | China | Retrospective | Posterior LIF/transforaminal LIF | ② | 367 | 20 | 7 |
| *Xu et al. (2024)* | 2024 | China | Prospective | Posterior LIF | ①②④ | 401 | 36 | 8 |

Note:
*AHRQ; ①Symptoms suspected to be SIJP, such as the lower lumbar and buttock pain below the L5 spinous process, postoperative pain differ from the preoperative one, with no evidence of lumbar cause, with or without sitting intolerance/difficulty turning around in bed, et al. ②At least two positive provocative tests of SIJ, such as Thigh thrust, Iliac distraction test, Gaenslen's test, Patrick's FABER test, sacral compression, Shear test, Yeoman maneuver, et al. ③A Diagnostic Scoring System for SIJP (*Kurosawa et al., 2017*), ranging from 0 to 9 points, had a cutoff value four. ④There is a positive response to SIJ injection/block/infiltration.

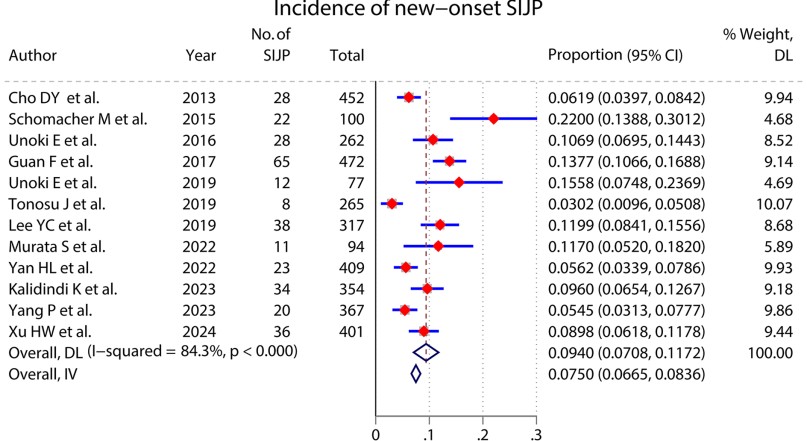

**Figure 2 Incidence of new-onset SIJP.**

effects model indicated that the incidence of new-onset SIJP after spinal surgery was 9.40% (95% CI [0.0708–0.1172]).

## Meta-analyses of risk factors (Figs. 4–9 and Table 2)

We performed meta-analyses for six factors, respectively. Data for some of these factors were subjected to subgroup analyses following dichotomization according to different

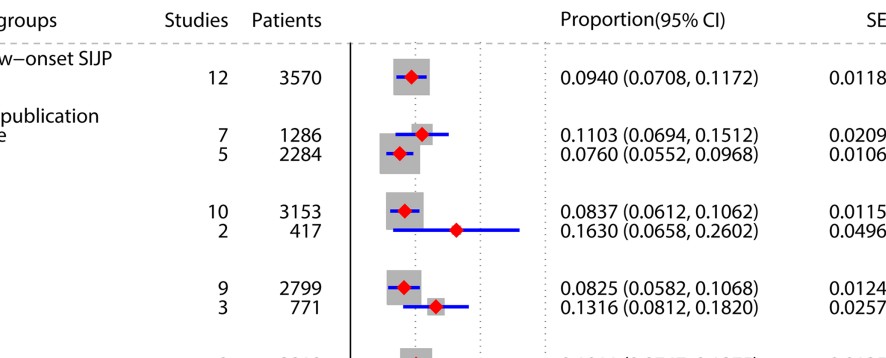

Figure 3 Subgroup for the incidence of new-onset SIJP after spinal surgery.

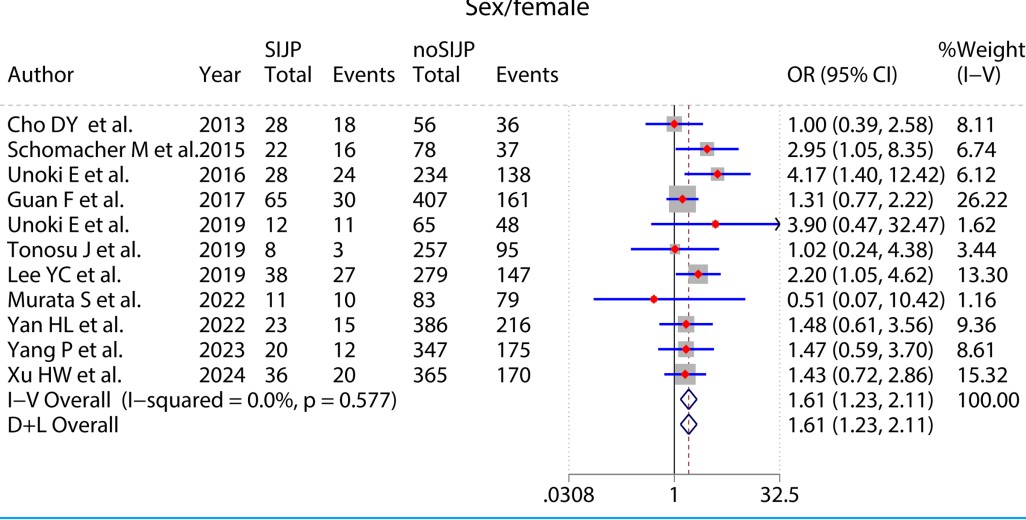

Figure 4 Forest plot for sex.

nodes, and we discarded the overall results of subgroup analyses for these factors. Each preoperative diagnosis was dichotomized by yes or no. Surgical segments were dichotomized by whether or not the surgical segments were equal or more than two, three, or four as the classification nodes, respectively. Spondylopelvic parameters in this study included preoperative and postoperative LL, PI, PT, and SS.

The sensitivity analyses and subgroup analyses were used to find sources of heterogeneity. The meta-analysis in the subgroup of surgical segments equal to or more

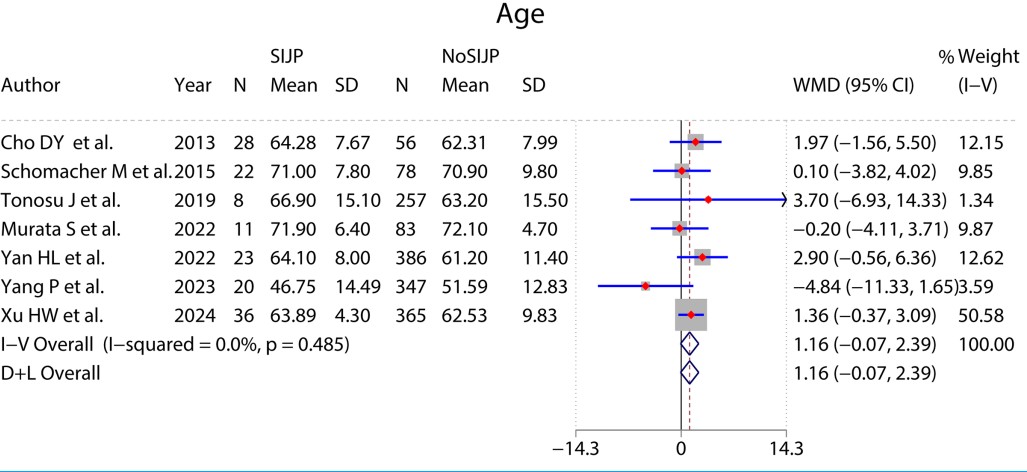

**Figure 5  Forest plot for age.**               

than four had moderate heterogeneity, which decreased from $I^2 = 53.1\%$ to $I^2 = 0\%$ by sensitivity analysis excluding *Lee, Lee & Harman (2019)*, and the statistical significance of the results did not change. The study was from Europe, whereas the other two were from Asia, which could be a potential reason for heterogeneity. We retained the study and used a random effects model for the meta-analysis (Fig. 7). The meta-analysis in lumbar disc herniation subgroup of preoperative diagnosis had moderate heterogeneity, which decreased from $I^2 = 50.1\%$ to $I^2 = 0\%$ by sensitivity analysis excluding *Xu et al. (2024)*. However, the statistical significance of the results would change, so we abandoned the quantitative analysis using a random effects model in favor of a qualitative systematic review. Four relevant studies all indicated no association between preoperative diagnosis of lumbar disc herniation and new-onset SIJP after spinal surgery (*Unoki et al., 2016*; *Guan et al., 2018*; *Tonosu et al., 2019*; *Xu et al., 2024*). Three of these studies showed a consistent trend that these patients had fewer new-onset SIJP after surgery, although without statistical significance. Meta-analyses of preoperative and postoperative LL both had high heterogeneity. By excluding the study by *Kalidindi et al. (2023)*, the heterogeneity of preoperative LL decreased from $I^2 = 89.2\%$ to $I^2 = 0\%$, with the heterogeneity of postoperative LL from $I^2 = 85.6\%$ to $I^2 = 39.7\%$, and the statistical significance of their results all did not change. The study by *Kalidindi et al. (2023)* was from India whereas all the other studies are from Asia. In addition, the mean number of surgical segments in the SIJP group was more than in the NoSIJP group for the study by *Kalidindi et al. (2023)* implying that more degenerating segments in the SIJP group resulted in a smaller LL; nevertheless, there was a similar mean number of surgical segments in the two groups for other studies, which could be potential causes for the difference in results. We therefore excluded the study and used a fixed effects model for meta-analyses of preoperative and postoperative LL (Fig. 10). The meta-analysis of postoperative PI had significant heterogeneity, which failed to find the source of heterogeneity by sensitivity and subgroup analysis; we used a random effects model to analyze.

# Preoperative diagnosis

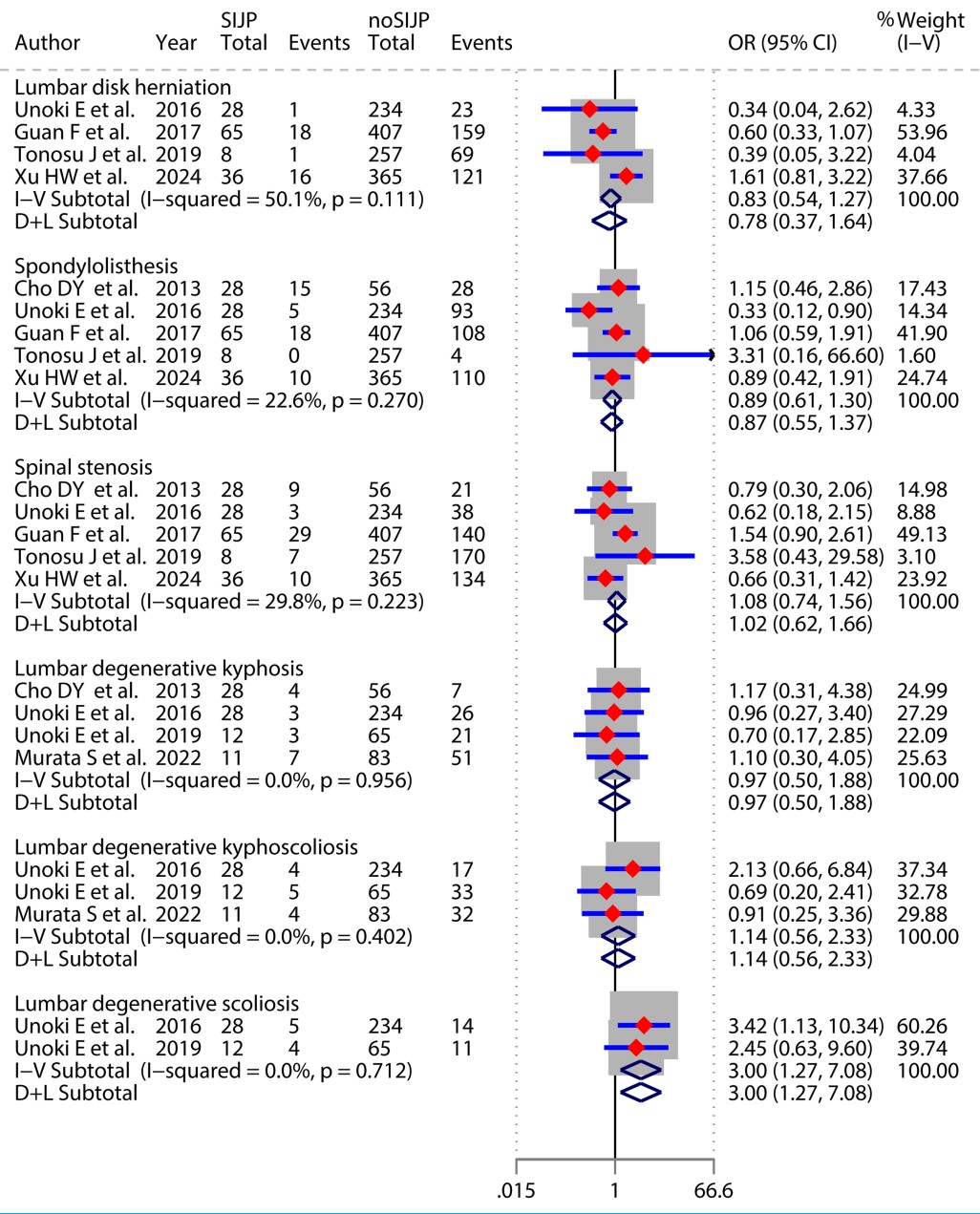

**Figure 6  Forest plot for preoperative diagnosis.**

The results of all meta-analyses were assessed for stability by sensitivity analysis, with qualitative systematic reviews instead of unstable quantitative analysis results. The sensitivity analysis showed that the meta-analysis results of age, preoperative diagnosis of lumbar degenerative scoliosis, preoperative PI and PT, and postoperative SS were unstable. All seven studies involving patients' age showed no significant difference in patient age between the SIJP and NoSIJP groups (*Cho et al., 2013*; *Schomacher et al., 2015*; *Tonosu et al., 2019*; *Murata et al., 2022*; *Yan et al., 2022*; *Yang et al., 2023*; *Xu et al., 2024*). The two

## No. of surgical segments

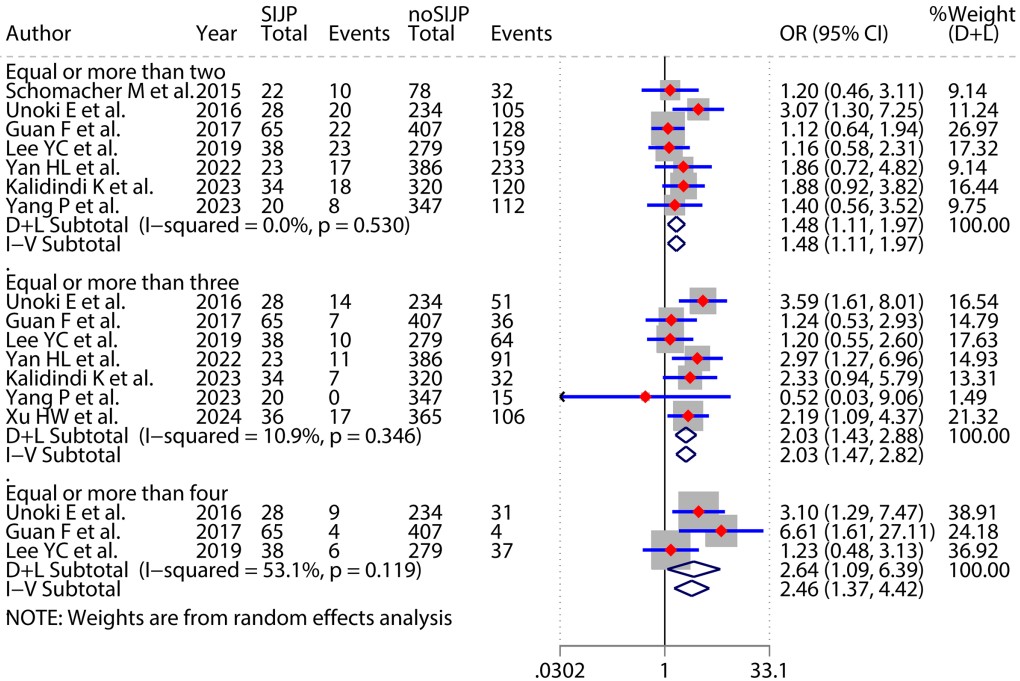

| Author | Year | SIJP Total | Events | noSIJP Total | Events | | OR (95% CI) | %Weight (D+L) |
|--------|------|------------|--------|--------------|--------|---|-------------|---------------|
| **Equal or more than two** | | | | | | | | |
| Schomacher M et al. | 2015 | 22 | 10 | 78 | 32 | | 1.20 (0.46, 3.11) | 9.14 |
| Unoki E et al. | 2016 | 28 | 20 | 234 | 105 | | 3.07 (1.30, 7.25) | 11.24 |
| Guan F et al. | 2017 | 65 | 22 | 407 | 128 | | 1.12 (0.64, 1.94) | 26.97 |
| Lee YC et al. | 2019 | 38 | 23 | 279 | 159 | | 1.16 (0.58, 2.31) | 17.32 |
| Yan HL et al. | 2022 | 23 | 17 | 386 | 233 | | 1.86 (0.72, 4.82) | 9.14 |
| Kalidindi K et al. | 2023 | 34 | 18 | 320 | 120 | | 1.88 (0.92, 3.82) | 16.44 |
| Yang P et al. | 2023 | 20 | 8 | 347 | 112 | | 1.40 (0.56, 3.52) | 9.75 |
| D+L Subtotal (I−squared = 0.0%, p = 0.530) | | | | | | | 1.48 (1.11, 1.97) | 100.00 |
| I−V Subtotal | | | | | | | 1.48 (1.11, 1.97) | |
| **Equal or more than three** | | | | | | | | |
| Unoki E et al. | 2016 | 28 | 14 | 234 | 51 | | 3.59 (1.61, 8.01) | 16.54 |
| Guan F et al. | 2017 | 65 | 7 | 407 | 36 | | 1.24 (0.53, 2.93) | 14.79 |
| Lee YC et al. | 2019 | 38 | 10 | 279 | 64 | | 1.20 (0.55, 2.60) | 17.63 |
| Yan HL et al. | 2022 | 23 | 11 | 386 | 91 | | 2.97 (1.27, 6.96) | 14.93 |
| Kalidindi K et al. | 2023 | 34 | 7 | 320 | 32 | | 2.33 (0.94, 5.79) | 13.31 |
| Yang P et al. | 2023 | 20 | 0 | 347 | 15 | | 0.52 (0.03, 9.06) | 1.49 |
| Xu HW et al. | 2024 | 36 | 17 | 365 | 106 | | 2.19 (1.09, 4.37) | 21.32 |
| D+L Subtotal (I−squared = 10.9%, p = 0.346) | | | | | | | 2.03 (1.43, 2.88) | 100.00 |
| I−V Subtotal | | | | | | | 2.03 (1.47, 2.82) | |
| **Equal or more than four** | | | | | | | | |
| Unoki E et al. | 2016 | 28 | 9 | 234 | 31 | | 3.10 (1.29, 7.47) | 38.91 |
| Guan F et al. | 2017 | 65 | 4 | 407 | 4 | | 6.61 (1.61, 27.11) | 24.18 |
| Lee YC et al. | 2019 | 38 | 6 | 279 | 37 | | 1.23 (0.48, 3.13) | 36.92 |
| D+L Subtotal (I−squared = 53.1%, p = 0.119) | | | | | | | 2.64 (1.09, 6.39) | 100.00 |
| I−V Subtotal | | | | | | | 2.46 (1.37, 4.42) | |

NOTE: Weights are from random effects analysis

.0302          1          33.1

**Figure 7 Forest plot for No. of surgical segment.**

## Fusion to sacrum

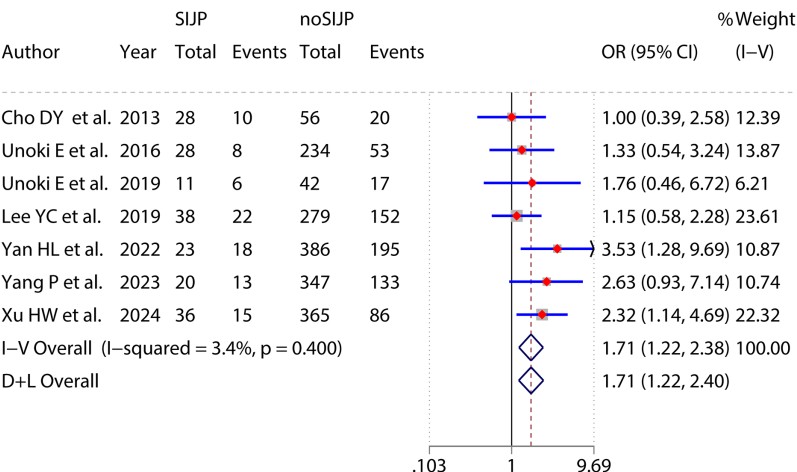

| Author | Year | SIJP Total | Events | noSIJP Total | Events | | OR (95% CI) | %Weight (I−V) |
|--------|------|------------|--------|--------------|--------|---|-------------|---------------|
| Cho DY et al. | 2013 | 28 | 10 | 56 | 20 | | 1.00 (0.39, 2.58) | 12.39 |
| Unoki E et al. | 2016 | 28 | 8 | 234 | 53 | | 1.33 (0.54, 3.24) | 13.87 |
| Unoki E et al. | 2019 | 11 | 6 | 42 | 17 | | 1.76 (0.46, 6.72) | 6.21 |
| Lee YC et al. | 2019 | 38 | 22 | 279 | 152 | | 1.15 (0.58, 2.28) | 23.61 |
| Yan HL et al. | 2022 | 23 | 18 | 386 | 195 | | 3.53 (1.28, 9.69) | 10.87 |
| Yang P et al. | 2023 | 20 | 13 | 347 | 133 | | 2.63 (0.93, 7.14) | 10.74 |
| Xu HW et al. | 2024 | 36 | 15 | 365 | 86 | | 2.32 (1.14, 4.69) | 22.32 |
| I−V Overall (I−squared = 3.4%, p = 0.400) | | | | | | | 1.71 (1.22, 2.38) | 100.00 |
| D+L Overall | | | | | | | 1.71 (1.22, 2.40) | |

.103          1          9.69

**Figure 8 Forest plot for fusion to sacrum.**

studies involving lumbar degenerative scoliosis showed no association between preoperative diagnosis of lumbar degenerative scoliosis and new-onset SIJP after spinal surgery (*Unoki et al., 2016*, *2019*); in contrast, the pooled results showed statistically significant but still need further studies by more studies. For preoperative PI, three studies had no significant difference between two groups (*Cho et al., 2013*; *Murata et al., 2022*; *Yang et al., 2023*), and other two showed that preoperative PI was significantly higher in
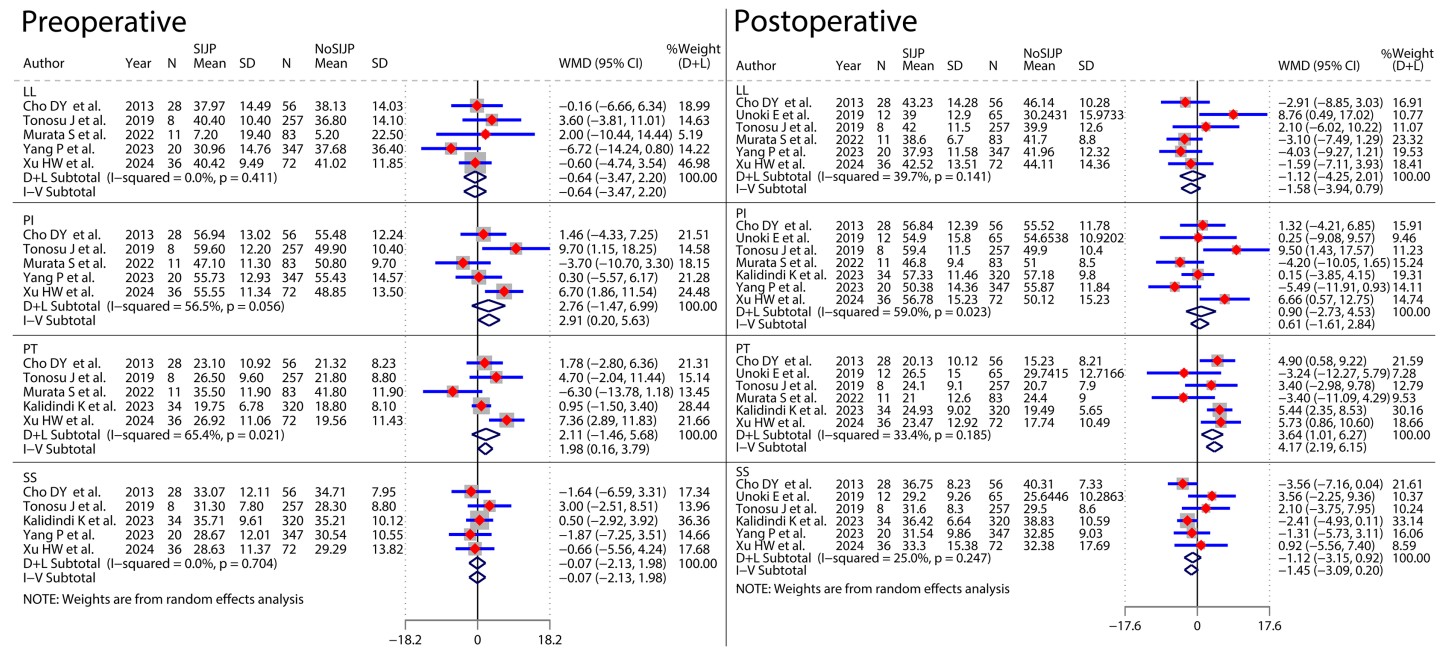

**Figure 9 Forest plot for two group controls of spondylopelvic parameters in patients.**

the SIJP group than in the NoSIJP group (*Tonosu et al., 2019*; *Xu et al., 2024*). For preoperative PT, four studies had no significant difference between two groups (*Cho et al., 2013*; *Tonosu et al., 2019*; *Murata et al., 2022*; *Kalidindi et al., 2023*), only one study showed that preoperative PI was significantly larger in the SIJP group than in the NoSIJP group (*Xu et al., 2024*). For postoperative SS, all six studies had no significant difference between two groups in postoperative SS (*Cho et al., 2013*; *Tonosu et al., 2019*; *Unoki et al., 2019*; *Kalidindi et al., 2023*; *Yang et al., 2023*; *Xu et al., 2024*).

We assessed publication bias for meta-analyses of sex, and the results were $P = 0.755$ for Begg's and $P = 0.666$ for Egger's tests, suggesting symmetry of funnel plots and no publication bias (Fig. 10).

The meta-analysis results of sex (Fig. 4), number of surgical segments (Fig. 7), fusion to sacrum (Fig. 8), postoperative PT (Fig. 9) showed statistically significant differences between the SIJP and the NoSIJP groups. All other factors had no significant difference between the SIJP and NoSIJP groups. See Table 2 for all results.

## Meta-analyses for pre- and postoperative controls of spondylopelvic parameters in patients (Fig. 11 and Table 3)

Most of the included studies did not perform pre- and postoperative control analyses of spondylopelvic parameters, and we attempted pre- and postoperative control meta-analysis after summarising and collating the data. Pre- and postoperative control meta-analyses of LL in SIJP or NoSIJP groups all had high heterogeneity, which decreased to 0 after excluding one study that was the heterogeneity source by sensitivity analysis (*Murata et al., 2022*). In this study, patients underwent multisegmental corrective fusion with significant changes in postoperative LL, whereas in other studies, patients underwent

**Table 2 Meta-analyses results for two group controls.**

| Risk factors | Subgroups | No of studies | No of patients | | Heterogeneity test | | Effect model | Meta-analysis results | | |
|---|---|---|---|---|---|---|---|---|---|---|
| | | | SIJP | No SIJP | P value | $I^2$(%) | | Effect size | 95% CI | P value |
| Sex | Female | 11 (*Cho et al., 2013*; *Schomacher et al., 2015*; *Unoki et al., 2016*; *Guan et al., 2018*; *Lee, Lee & Harman, 2019*; *Tonosu et al., 2019*; *Unoki et al., 2019*; *Murata et al., 2022*; *Yan et al., 2022*; *Yang et al., 2023*; *Xu et al., 2024*) | 291 | 2,557 | 0.58 | 0.0 | FEM | OR = 1.61 | [1.23–2.11] | 0.001 |
| Age* | | 7 (*Cho et al., 2013*; *Schomacher et al., 2015*; *Tonosu et al., 2019*; *Murata et al., 2022*; *Yan et al., 2022*; *Yang et al., 2023*; *Xu et al., 2024*) | 148 | 1,572 | 0.49 | 0.0 | FEM | WMD = 1.16 | [−0.07 to 2.39] | 0.065 |
| Preoperative diagnosis | Lumbar disc herniation* | 4 (*Unoki et al., 2016*; *Guan et al., 2018*; *Tonosu et al., 2019*; *Xu et al., 2024*) | 137 | 1,263 | 0.11 | 50.1 | REM | OR = 0.78 | [0.37–1.64] | 0.516 |
| | Spondylolisthesis | 5 (*Cho et al., 2013*; *Unoki et al., 2016*; *Guan et al., 2018*; *Tonosu et al., 2019*; *Xu et al., 2024*) | 165 | 1,319 | 0.27 | 22.6 | FEM | OR = 0.89 | [0.61–1.30] | 0.539 |
| | Spinal stenosis | 5 (*Cho et al., 2013*; *Unoki et al., 2016*; *Guan et al., 2018*; *Tonosu et al., 2019*; *Xu et al., 2024*) | 165 | 1,319 | 0.23 | 29.8 | FEM | OR = 1.08 | [0.74–1.56] | 0.696 |
| | Degenerative kyphosis | 4 (*Cho et al., 2013*; *Unoki et al., 2016, 2019*; *Murata et al., 2022*) | 79 | 438 | 0.96 | 0.0 | FEM | OR = 0.97 | [0.50–1.88] | 0.934 |
| | Degenerative kyphoscoliosis | 3 (*Unoki et al., 2016, 2019*; *Murata et al., 2022*) | 51 | 382 | 0.40 | 0.0 | FEM | OR = 1.14 | [0.56–2.33] | 0.714 |
| | Degenerative scoliosis* | 2 (*Unoki et al., 2016, 2019*) | 40 | 299 | 0.71 | 0.0 | FEM | OR = 3.00 | [1.27–7.08] | 0.012 |
| No. of surgical segments | ≥2 | 7 (*Schomacher et al., 2015*; *Unoki et al., 2016*; *Guan et al., 2018*; *Lee, Lee & Harman, 2019*; *Yan et al., 2022*; *Kalidindi et al., 2023*; *Yang et al., 2023*) | 230 | 2,051 | 0.53 | 0.0 | FEM | OR = 1.48 | [1.11–1.97] | 0.008 |
| | ≥3 | 7 (*Unoki et al., 2016*; *Guan et al., 2018*; *Lee, Lee & Harman, 2019*; *Yan et al., 2022*; *Kalidindi et al., 2023*; *Yang et al., 2023*; *Xu et al., 2024*) | 244 | 2,338 | 0.35 | 10.9 | FEM | OR = 2.03 | [1.47–2.82] | 0.000 |
| | ≥4 | 3 (*Unoki et al., 2016*; *Guan et al., 2018*; *Lee, Lee & Harman, 2019*) | 131 | 920 | 0.12 | 53.1 | REM | OR = 2.64 | [1.09–6.39] | 0.031 |
| Fusion to sacrum | | 7 (*Cho et al., 2013*; *Unoki et al., 2016*; *Lee, Lee & Harman, 2019*; *Unoki et al., 2019*; *Yan et al., 2022*; *Yang et al., 2023*; *Xu et al., 2024*) | 184 | 1,709 | 0.40 | 3.4 | FEM | OR = 1.71 | [1.22–2.38] | 0.002 |

| Risk factors | Subgroups | No of studies | No of patients | | Heterogeneity test | | Effect model | Meta-analysis results | | |
|---|---|---|---|---|---|---|---|---|---|---|
| | | | SIJP | No SIJP | P value | $I^2$(%) | | Effect size | 95% CI | P value |
| Spondylopelvic parameters | Preoperative LL[#] | 6 (*Cho et al., 2013*; *Tonosu et al., 2019*; *Murata et al., 2022*; *Kalidindi et al., 2023*; *Yang et al., 2023*; *Xu et al., 2024*)-1 (*Kalidindi et al., 2023*) | 103 | 815 | 0.41 | 0.0 | FEM | WMD = −0.64 | [−3.47 to 2.20] | 0.660 |
| | Postoperative LL[#] | 7 (*Cho et al., 2013*; *Tonosu et al., 2019*; *Unoki et al., 2019*; *Murata et al., 2022*; *Kalidindi et al., 2023*; *Yang et al., 2023*; *Xu et al., 2024*)-1(*Kalidindi et al., 2023*) | 115 | 880 | 0.14 | 39.7 | FEM | WMD = −1.58 | [−3.94 to 0.79] | 0.191 |
| | Preoperative PI* | 5 (*Cho et al., 2013*; *Tonosu et al., 2019*; *Murata et al., 2022*; *Yang et al., 2023*; *Xu et al., 2024*) | 103 | 815 | 0.06 | 56.5 | REM | WMD = 2.76 | [−1.47 to 6.99] | 0.201 |
| | Postoperative PI | 7 (*Cho et al., 2013*; *Tonosu et al., 2019*; *Unoki et al., 2019*; *Murata et al., 2022*; *Kalidindi et al., 2023*; *Yang et al., 2023*; *Xu et al., 2024*) | 149 | 1,200 | 0.02 | 59.0 | REM | WMD = 0.90 | [−2.73 to 4.53] | 0.628 |
| | Preoperative PT* | 5 (*Cho et al., 2013*; *Tonosu et al., 2019*; *Murata et al., 2022*; *Kalidindi et al., 2023*; *Xu et al., 2024*) | 117 | 788 | 0.02 | 65.4 | REM | WMD = 2.11 | [−1.46 to 5.68] | 0.247 |
| | Postoperative PT | 6 (*Cho et al., 2013*; *Tonosu et al., 2019*; *Unoki et al., 2019*; *Murata et al., 2022*; *Kalidindi et al., 2023*; *Xu et al., 2024*) | 129 | 853 | 0.19 | 33.4 | FEM | WMD = 4.17 | [2.19–6.15] | 0.000 |
| | Preoperative SS | 5 (*Cho et al., 2013*; *Tonosu et al., 2019*; *Kalidindi et al., 2023*; *Yang et al., 2023*; *Xu et al., 2024*) | 126 | 1,052 | 0.70 | 0.0 | FEM | WMD = −0.07 | [−2.13 to 1.98] | 0.944 |
| | Postoperative SS* | 6 (*Cho et al., 2013*; *Tonosu et al., 2019*; *Unoki et al., 2019*; *Kalidindi et al., 2023*; *Yang et al., 2023*; *Xu et al., 2024*) | 138 | 1,117 | 0.25 | 25.0 | FEM | WMD = −1.45 | [−3.09 to 0.20] | 0.085 |

**Note:**
①[#]The source of heterogeneity was found and excluded by sensitivity analysis; *The results were assessed as unstable by sensitivity analysis, with qualitative systematic reviews instead of unstable quantitative analysis results. ②FEM: fixed effects model; REM: random effects model.

short segmental fusion or decompression, thus generating heterogeneity. After excluding this study, the results of pre- and postoperative LL control meta-analyses in both SIJP or No SIJP groups were stable, and we retained this study and used random effects models. Pre- and postoperative control meta-analysis of PT in SIJP or NoSIJP groups all had significant heterogeneity, and sensitivity analysis suggested that analysis results were unstable. Instead of using the overall results of quantitative analyses, we used qualitative

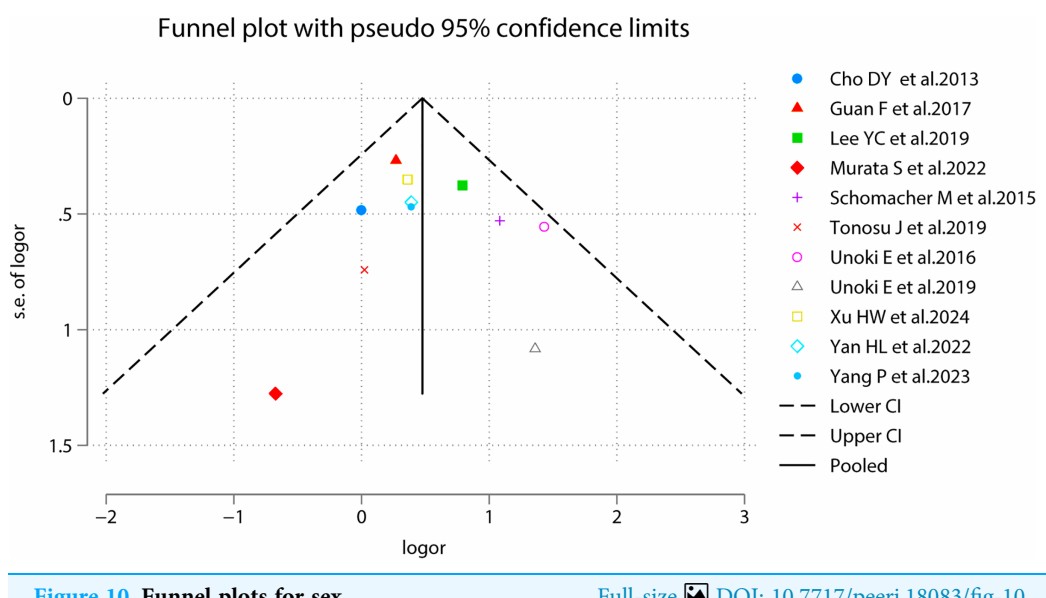

**Figure 10  Funnel plots for sex.**

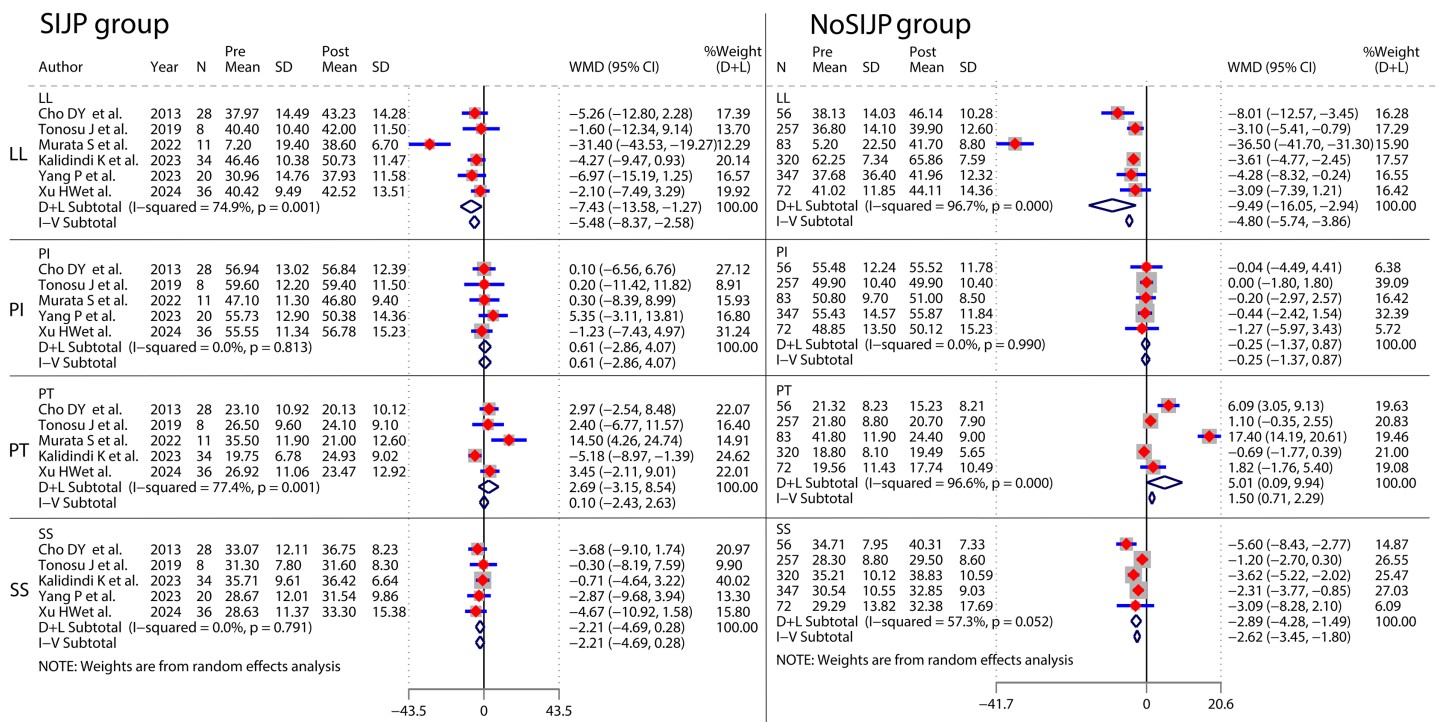

**Figure 11  Forest plot for pre- and postoperative control meta-analyses of spondylopelvic parameters.**

systematic reviews to describe the results shown in the forest plot of the meta-analysis. For the SIJP group, three studies had no significant change (*Cho et al., 2013*; *Tonosu et al., 2019*; *Xu et al., 2024*), whereas the other two had a significant decrease in postoperative PT of patients compared to preoperative (*Murata et al., 2022*; *Kalidindi et al., 2023*). For the

**Table 3 Meta-analysis results for pre- and postoperative controls of spondylopelvic parameters in patients.**

| Spondylopelvic parameters | Patients with SIJP/ NoSIJP | No of studies | No of patients | Heterogeneity test | | Effect model | Meta-analysis results | | |
|---|---|---|---|---|---|---|---|---|---|
| | | | | *P* value | $I^2$(%) | | WMD | 95% CI | *P* value |
| LL | SIJP | 6 (*Cho et al., 2013*; *Tonosu et al., 2019*; *Murata et al., 2022*; *Kalidindi et al., 2023*; *Yang et al., 2023*; *Xu et al., 2024*) | 137 | 0.001 | 74.9 | REM | −7.43 | [−13.58 to −1.27] | 0.018 |
| | NoSIJP | | 1,135 | 0.00 | 96.7 | REM | −9.49 | [−16.05 to −2.94] | 0.005 |
| PI | SIJP | 5 (*Cho et al., 2013*; *Tonosu et al., 2019*; *Murata et al., 2022*; *Yang et al., 2023*; *Xu et al., 2024*) | 103 | 0.81 | 0.0 | FEM | 0.61 | [−2.86 to 4.07] | 0.731 |
| | NoSIJP | | 815 | 0.99 | 0.0 | FEM | −0.25 | [−1.37 to 0.87] | 0.662 |
| PT | SIJP* | 5 (*Cho et al., 2013*; *Tonosu et al., 2019*; *Murata et al., 2022*; *Kalidindi et al., 2023*; *Xu et al., 2024*) | 117 | 0.001 | 77.4 | REM | 2.69 | [−3.15 to 8.54] | 0.366 |
| | NoSIJP* | | 788 | 0.00 | 96.6 | REM | 5.01 | [0.09–9.94] | 0.046 |
| SS | SIJP | 5 (*Cho et al., 2013*; *Tonosu et al., 2019*; *Kalidindi et al., 2023*; *Yang et al., 2023*; *Xu et al., 2024*) | 126 | 0.80 | 0.0 | FEM | −2.21 | [−4.69 to 0.28] | 0.082 |
| | NoSIJP | | 1,052 | 0.05 | 57.3 | REM | −2.89 | [−4.28 to −1.49] | 0.000 |

**Note:**
①*The results were assessed as unstable by sensitivity analysis, with qualitative systematic reviews instead of unstable quantitative analysis results. ②FEM: fixed effects model; REM: random effects model.

NoSIJP group, three studies had no significant change (*Tonosu et al., 2019*; *Kalidindi et al., 2023*; *Xu et al., 2024*), whereas the other two had a significant decrease in postoperative PT of patients compared to preoperative (*Cho et al., 2013*; *Murata et al., 2022*). Pre- and postoperative control meta-analyses of SS in the NoSIJP group had moderate heterogeneity, which failed to find the source of heterogeneity by sensitivity and subgroup analysis; we used a random effects model to analyze. Postoperative LL in the SIJP group and postoperative LL and SS of patients in the NoSIJP group had significant differences from preoperative. All other postoperative spondylopelvic parameters had no significant difference with preoperative, regardless of the patients with SIJP or NoSIJP. See Table 3 for all results.

## DISCUSSION

Spinal surgeries, mainly on the lumbar spine, are one of the major causes contributing to SIJP. The pain may be due to surgery-induced increased SIJ stress load, heightened range of motion, and damage to surrounding nerve tissue (*Ivanov et al., 2009*; *Yoshihara, 2012*). The meta-analysis results in this study indicated that the incidence of SIJP after spinal surgery was approximately 9.40%, with a high inter-study heterogeneity among the included studies. Subgroup analysis did not find the source of significant heterogeneity, which may be closely related to the type of surgery, and the diagnostic standard was not fully consistent across studies could also be one of the essential reasons.

Regarding whether sex is a potential risk factor for new-onset SIJP after spinal surgery, the included studies did not pay enough attention to it, and most of them were described only in the baseline data without statistical analysis. We collected and collated the data and then performed the meta-analysis, which indicated that females had a higher risk for
new-onset SIJP after spinal surgery. The SIJ anatomy exhibits a distinct sexual dimorphism, with females having a slightly smaller surface area of the SIJ, a shorter cylindrical pelvic cavity, and a relatively wider, more uneven, less curved, and more posterior inclination (*Ulas, Diekhoff & Ziegeler, 2023*). Differences in anatomy contribute to the distinct biomechanics of the SIJ, with higher mobility, stress, load, and pelvic ligament strain in females SIJ compared to males, resulting in greater stress on the entire joint and higher rates of joint misalignment (*Joukar et al., 2018*). There are also gender differences in the movement pattern of sagittal rotation for SIJ, such as females from a supine to a standing position with a greater posterior rotation of the ilium relative to the sacrum than males (*Tani et al., 2023*); the SIJ motion during trunk extension is also significantly greater in females than males for patients with degenerative lumbar spine disorders (DLSDs) (*Nagamoto et al., 2015*). In addition, the ligamentous complex of the female SIJ is more flexible due to reproductive physiological needs, and factors during pregnancy, such as weight gain, lordotic posture, hormone-induced ligamentous laxity in late pregnancy, and pelvic trauma, can induce or even exacerbate SIJP (*Gutke, Ostgaard & Oberg, 2006*). Studies by *Tonosu et al. (2021)* and *DePalma, Ketchum & Saullo (2012)* also identified females as a risk factor for primary SIJP. However, the female patients in this study did not have preoperative SIJP but had new-onset SIJP after spinal surgery. The primary reason for this may be that spinal surgery unavoidably destroys the muscular ligamentous tissues of the lower back, exacerbating the instability of the female SIJ.

A study showed that compared to patients who had only symptomatic low back pain, patients with lumbar disc herniation have more severe SIJ degeneration with more pathological changes, symptoms, and complications (*Huang et al., 2021*). Although none of the included studies showed a statistically significant association between preoperative diagnosis of lumbar disc herniation and new-onset SIJP after spinal surgery, we cannot ignore the trend that patients with lumbar disc herniation in these studies have fewer new-onset SIJP after surgery (*Unoki et al., 2016*; *Guan et al., 2018*; *Tonosu et al., 2019*). The trend perhaps meant that SIJ degeneration of patients with lumbar disc herniation is not too bad compared to other patients with severe degenerative spinal diseases who require or have undergone surgical treatment. However, a randomized controlled trial of adult spinal deformity (ASD) patients showing a prevalence of already 16% of SIJP before surgery suggests that ASD patients may be susceptible to SIJP (*Polly et al., 2024*). Furthermore, the main reason for the differences may be that patients with lumbar disc herniation can undergo decompression surgery or fusion surgery, and patients with other severe degenerative spinal diseases mostly undergo lumbar fusion or even multiple-segment corrective fusion. *Guan et al. (2018)* also showed a higher incidence of new-onset SIJP after surgery in patients undergoing posterior lumbar interbody fusion (PLIF) than in patients undergoing primary lumbar open discectomy.

The results of our analyses are consistent with the majority view that fusion to the sacrum is a significant factor contributing to new-onset SIJP after surgery in patients with spinal fusion (*Yan et al., 2022*; *Manzetti et al., 2023*; *Shen et al., 2023*; *Yang et al., 2023*). A possible cause of SIJP after fusion to the sacrum is some unavoidable damage to the ligaments, muscles, and even neural tissues surrounding the SIJ during surgical procedures

with the insertion of the pedicle screws and the fixation process of connecting rods (*Yan et al., 2022*). Additionally, the stresses of internal fixation will directly affect the SIJ and its surrounding vital anatomical structures, thus increasing the sacrum angular motion of the SIJ, average stress on the SIJ articular surface, and maximum strain in the iliosacral ligament and ileal ligament, which may be a primary factor contributing to SIJP (*Ivanov et al., 2009*; *Yao et al., 2023*).

Multi-segment surgery is a risk factor for new-onset SIJP after spinal surgery, with an increasing risk associated with the number of surgical segments. The number of fused segments is one of the critical risk factors for adjacent segment degeneration of L5/S1 after L5 floating lumbar fusion (*Takegami et al., 2023*). Biomechanical studies also showed that as the number of spinal fixation segments increases, the range of motion and intradiscal pressure at both adjacent and distal segments also increase (*Nagata et al., 1993*; *Mu et al., 2019*; *Ou et al., 2021*). Although the above studies did not directly assess SIJ, SIJ as an adjacent joint to the lumbar spine also showed similar biomechanical behaviors after lumbar/lumbosacral fusion surgery (*Ivanov et al., 2009*), which leads to new-onset SIJP after surgery. Indeed, SIJP may also be associated with excessive disruption of anatomical structures such as muscle and ligaments or significant reconstruction of sagittal balance by multisegmental surgery (*Yan et al., 2022*). A study had shown that patients who underwent long spinal fusion had larger LL and smaller SS after surgery than those who underwent short spinal fusion. Their sagittal balance was improved, but there was still more pelvic retroversion, which may increase the range of motion of SIJ (*Ukai et al., 2023*). *Unoki et al. (2016)* and *Ackerman, Deol & Polly (2022)* advocate the addition of pelvic fixation/fusion in multi-segment fusion surgery, providing stress support through screws and reducing SIJ mobility (*de Andrada Pereira et al., 2022*), thereby decreasing loads on the SIJ and preventing SIJP (*Volkheimer et al., 2017*; *Mushlin et al., 2019*), because patients with pelvic fixation compensated for sagittal imbalance more through flexion of the knee rather than the movement of the SIJ (*Zhi et al., 2023*).

Measuring radiographic spondylopelvic parameters to assess sagittal balance has become increasingly crucial in spinal surgery (*Le Huec et al., 2015*). Numerous studies have explored the relationship between sagittal balance and various spinal degenerative changes (*Barrey et al., 2007*; *Diebo et al., 2019*; *Thornley et al., 2023*). A study by *Kwon et al. (2020)* found that compared with patients with lumbar spinal stenosis without sagittal imbalance, adult spinal deformity patients with spinal imbalance show more serious SIJ degeneration. There is an association between new-onset SIJP after spinal surgery and spinopelvic parameters. This study had no significant differences in preoperative LL, PI, PT, and SS between the SIJP and the NoSIJP groups, which implies that we cannot predict whether patients will have new-onset SIJP after surgery based on preoperative parameters. Moreover, patients with new-onset SIJP had postoperatively smaller SS and significantly larger PT than patients without new-onset SIJP. Increased PT and decreased SS represent pelvic retroversion, one of the main compensatory mechanisms for maintaining sagittal balance (*Barrey et al., 2013*). *Tchachoua Jiembou, Nda & Konan (2023)* indicated that persistent pelvic retroversion after surgery was indicative of an arthrodesis performed on an unbalanced spine. Thus, our results showed that patients with SIJP still had more pelvic

retroversion and persistent sagittal imbalance after surgery. This is consistent with our pooled results regarding preoperative and postoperative control of spinopelvic parameters that patients in the NoSIJP group had a significant increase in postoperative SS compared to preoperative, while the increase in the SIJP group was not statistically significant. A biomechanical study showed that the L5-S1 range of motion and intervertebral disc pressure (IDP) gradually decreased with the increase of SS after L4-L5 lumbar fusion (*Ke et al., 2020*), which decreases the risk of L5-S1 intervertebral disc degeneration, and pressure on the SIJ may also relieve accordingly (*Sato et al., 2020*). The increase in SS also implies that patients in the NoSIJP group had decreased postoperative pelvic retroversion compensation. Pelvic retroversion was an essential cause for new-onset SIJP after spinal surgery because pelvic retroversion activates the gravity line back, producing a backward lever arm on the sacroiliac joint through the femoral head and transmitting reaction forces from the ground, then causing twisting mobilization of SIJ and severe destruction of the surrounding complex ligament structures *(Jean, 2014)*. Based on the positive correlation between SS and LL (*Roussouly et al., 2005*), postoperative LL of patients also had significant improvement compared to preoperative in both groups. Many studies have confirmed that the correction of LL can improve the sagittal balance and reduce complications after spinal surgery, such as adjacent segment degeneration or proximal junctional kyphosis (*Lee et al., 2016*; *Im et al., 2020*; *Wang et al., 2020*). However, the relationship between LL and new-onset SIJP after spinal surgery remains to be explored. Surgical intervention usually does not directly change the pelvis, when pelvic compensation is no longer necessary due to the ideal correction of LL, PT will improve with the hip reversion from the terminal extension and the pelvis forward rotation (*Zhang et al., 2021*), and SS increases correspondingly due to a geometrical relationship (PI = PT + SS (*Legaye et al., 1998*)). Theoretically, postoperative PT might significantly decrease with a significant increase in postoperative SS of patients in the NoSIJP group; however, this speculation was not confirmed in this study due to the unstable results of postoperative PT. Based on the available results, we speculated that patients in the SIJP group did not achieve better sagittal balance after surgery; in contrast, patients in the NoSIJP group showed a significant improvement in sagittal balance after surgery, although we did not directly compare the improvement before and after surgery between the two groups.

## STRENGTHS AND LIMITATIONS

The main strength of this study lies in being the first meta-analysis on the incidence and potential risk factors of new-onset SIJP after spinal surgery. Furthermore, we did not limit our meta-analysis to only the relevant factors already analyzed in the included studies. Instead, based on available data extracted from the included studies, we performed additional control analyses yet to be conducted by the researchers in included studies, such as patients' preoperative and postoperative controls, which provided direction for subsequent studies. Of course, this study also has some limitations. Firstly, the number of included studies was limited; some indicators also needed more relevant studies to report; meanwhile, most included studies were retrospective observational studies, and fewer had the results of multivariate regression analyses. Second, due to various reasons, such as

differences in study types and population, sources of heterogeneity were still unidentified despite conducting the subgroup analyses. In addition, this study included only English-language literature without articles published in other languages in this field, which potentially introduces bias. Further exploration of the risk factors for new-onset SIJP after spinal surgery through high-quality, large-sample prospective clinical cohort studies by more scholars is necessary.

## CONCLUSIONS

In summary, the incidence of new-onset SIJP after spinal surgery is approximately 9.40%. The results of this study showed that sex, multi-segmental surgery, and fusion to the sacrum were risk factors for new-onset SIJP after spinal surgery. Larger postoperative PT increases the risk of new-onset SIJP after spinal surgery. Surgeons should pay attention to the appropriate reconstruction of LL, reduction of compensatory pelvic retroversion, and improvement of sagittal balance during spinal surgery to reduce the occurrence of SIJP.

### Funding

This review was sponsored by the Traditional Chinese Medicine Inheritance and Innovative Talent Project (Zhongjing Project, No grant number) and the Henan Medical Association Second Batch of Grassroots Appropriate Technology Promotion Activity projects (No. SYJS2020067). The funders had no role in study design, data collection and analysis, decision to publish, or preparation of the manuscript.

### Grant Disclosures

The following grant information was disclosed by the authors:
Traditional Chinese Medicine Inheritance and Innovative Talent.
Henan Medical Association: No. SYJS2020067.

### Competing Interests

The authors declare that they have no competing interests.

### Author Contributions

- ChengHan Xu conceived and designed the experiments, performed the experiments, analyzed the data, prepared figures and/or tables, authored or reviewed drafts of the article, and approved the final draft.
- Xuxin Lin performed the experiments, analyzed the data, prepared figures and/or tables, authored or reviewed drafts of the article, and approved the final draft.
- Yingjie Zhou conceived and designed the experiments, authored or reviewed drafts of the article, and approved the final draft.
- Hanjie Zhuo conceived and designed the experiments, authored or reviewed drafts of the article, and approved the final draft.
- Lei Yang performed the experiments, authored or reviewed drafts of the article, and approved the final draft.

- Xubin Chai analyzed the data, authored or reviewed drafts of the article, and approved the final draft.
- Yong Huang analyzed the data, authored or reviewed drafts of the article, and approved the final draft.

## Data Availability

The raw data and code are available in the Supplemental Files.

## Supplemental Information

Supplemental information for this article can be found online at http://dx.doi.org/10.7717/peerj.18083#supplemental-information.

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
