# Peer review of "Incidence and risk factors of new-onset sacroiliac joint pain after spinal surgery: a systematic review and meta-analysis"

_PeerJ, doi:10.7717/peerj.18083_

## Round 0.1 · original submission · Major Revisions

· Academic Editor

Major Revisions

After reviewing the reviewers' comments, I have decided that the manuscript requires a major revision. The first reviewer provided positive feedback, suggesting only the removal of the '?' symbol in the discussion section. The second reviewer also praised the study but offered detailed suggestions to improve the discussion section, such as addressing Ivanov's findings on the effects of L5-S1 versus L4-S1 fusion, contrasting the statistics on sacroiliac (SI) joint pain with Manzetti's data, and ensuring uniformity in diagnosing SI joint pain using consistent definitions and clinical rules. Additionally, they suggested discussing SI joint pain in patients with adult spinal deformity (ASD) and its relation to pelvic retroversion, as well as the effects of pelvic fixation and compensation through hips and knees. The third reviewer raised a significant concern about the comprehensiveness of the literature search, highlighting the omission of a key article by Nessim A et al. (2021). They recommended that the search keywords and publication period be described in the methods section, and suggested including the missing references and re-running the meta-analysis. Therefore, the decision is to request a thorough revision, addressing the reviewers' suggestions, adjusting the literature search as needed, and ensuring that the meta-analysis is conducted comprehensively and accurately.

Reviewer 1 ·

Basic reporting

Well performed study. No comments regarding language use or data use. However, I would suggest not using '?' in the discussion section (line 290)
The conclusion is self-contained and correctly formed.

Experimental design

Sufficient description of methods design.

Validity of the findings

Findings are valid and well powered, complete data is available of several plots.

Additional comments

Well performed study on a relevant problem regarding SIJ dysfunction following lumbar spinal surgery. The findings are relevant and contribute to the scientific literature.

·

Basic reporting

Appropriate. Registered with PROSPERO and follows PRISMA guidelines.

Experimental design

As above- done appropriately.

Validity of the findings

They have nuanced out some new ideas that make sense but have not been reported yet in aggregate. See more details below.

Additional comments

The only opportunities for improvement are in their discussion section.

Line 264-5- Ivanov whom they have cited, noted a difference in the effect of fusion L5-S1 with a 54% increase in motion versus L4-S1 fusion which had a 160% increase in SI motion in their modeling. This leads one to believe that longer fusions create greater load/strain on the SI joint. Whether or not the clinical data is capable of discerning this difference is a challenging scenario.
Line 265-6 SI joint pain after fusion 9.67%. This should be contrasted with Manzetti where they found a weighted average of 24%.
Line 268-9 No uniform diagnosis of SI joint pain. The 2 randomized controlled trials of operative versus non-operative treatment and prospective cohorts have used the same definitions. 3 of 5 positive physical exam maneuvers (Thigh thrust, FABER, pelvic gapping, pelvic compression, Gaenslen’s) which has a 85% positive predictive value for a positive image guided injection response (Laslett) and is good enough to constitute a clinical diagnostic rule (Petersen BMC Musculoskeletal Disorders 2017).
Line 306-7 In patients with adult spinal deformity(ASD) enrolled in a RCT comparing fusion with or without concomitant SI fusion showed a prevalence of 16% of SI pain prior to surgery. ASD may have a predisposing diathesis to having SI joint pain. As the authors subsequently discuss this maybe a reflection of pelvic tilt/pelvic retroversion. (Polly et al World Neurosurgery 2024).
Line 329-332 insightful comments about the improved LL and decreased SS but still more pelvic retroversion.
Line 336 patients with pelvic fixation compensate more through hips and knees – again insightful comment.
Interestingly there is suggestive evidence that higher PI patients do not do as well in spinal fusion than lower PI patients. The effects of high PI on SI joint biomechanics is an intriguing question.
Again this is a well done manuscript following the appropriate methodology. The figures are well done and convey the data effectively.

Reviewer 3 ·

Basic reporting

Authors conducted a systemic review and meta-analyses on "incidence and risk factors of newly onset SIJ pain after spine surgery".

Experimental design

Overall the quality of paper is good, however, I have a major fundamental concern.

Validity of the findings

What were the key words used to search those literature? To my knowledge, one of the most important milestone article regarding this issue is "Infra-adjacent Segment Disease After Lumbar Fusion: An Analysis of Pelvic Parameters. Nessim A et al. Spine (Phila Pa 1976). 2021 Aug 15;46(16):E888-E892. doi: 10.1097/BRS.0000000000003998". How is it possible it was missed in this search?

Additional comments

Please describe the key words used to search the literature in the method section. In addition, the searching period of publication needs to be mentioned. Please pay attention to the references cited in the article I mentioned above, and consider to include any missing references, then please re-run the meta-analysis.

---

## Round 0.2 · Minor Revisions

· Academic Editor

Minor Revisions

Dear author, please see the sugestion from reviewer#3 and send to us a revised version.

Reviewer 3 ·

Basic reporting

n/a

Experimental design

n/a

Validity of the findings

n/a

Additional comments

Authors addressed my comments successfully except that Nessim's article was not cited in the reference list while they say it was not missed. Please update the reference list.

---

## Round 0.3 · accepted · Accept

· Academic Editor

Accept

After a thorough review of the revised manuscript, I am pleased to confirm that all of the reviewers' comments and suggestions have been carefully considered and adequately addressed. Although the previous reviewers were not invited to reassess the revised version, I have personally conducted a evaluation of the revisions and am satisfied with the changes made. The manuscript is now clear, coherent, and fully meets the quality standards required by our journal.
I am confident that the manuscript is ready for publication.